# PRINCIPAL TRADE-OFF ANALYSIS

## ABSTRACT

The focus on equilibrium solutions in games underemphasizes the importance of understanding their overall structure. A different set of tools is needed for learning and representing the general structure of a game. In this paper we illustrate "Principle Trade-off Analysis" (PTA), a decomposition method that embeds games into a low dimensional feature space and argue that the embeddings are more revealing than previously demonstrated. Here, we develop an analogy to Principal Component Analysis (PCA). PTA represents an arbitrary two-player zero-sum game as the weighted sum of pairs of orthogonal 2D feature planes. We show that each of the feature planes represent unique strategic trade-offs (cyclic modes) and truncation of the sequence provides insightful model reduction. We demonstrate the validity of PTA on a pair of games (Blotto, Pokemon). In Blotto, PTA identifies game symmetries, and specifies strategic trade-offs associated with distinct win conditions. These symmetries reveal limitations of PTA unaddressed in previous work. For Pokemon, PTA recovers clusters that naturally correspond to Pokemon types, correctly identifies the designed tradeoff between those types, and discovers a rock-paper-scissor (RPS) cycle in the Pokemon generation type - all absent any specific information except game outcomes.

## 1 INTRODUCTION

In recent years algorithms have achieved superhuman performance in a number of complex games such as Chess, Go, Shogi, Poker and Starcraft (Silver et al., 2018; Heinrich & Silver, 2016; Moravčík et al., 2017; Vinyals et al., 2019). Despite impressive game play, enhanced understanding of the game is typically only achieved by additional analysis of the algorithms game play post facto (Silver, 2018). Current work emphasizes the "policy problem", developing strong agents, despite growing demand for a task theory which addresses the "problem problem", i.e. what games are worth study and play (Omidshafiei et al., 2020; Clune, 2019). A task theory requires a language that characterizes and categorizes games, namely, a toolset of measures and visualization techniques that evaluate and illustrate game structure. Summary visuals and measures are especially important for complex games where direct analysis is intractable. In this vain tournaments are used to sample the game and to empirically evaluate agents. The empirical analysis of tournaments has a long history, in sports analytics (Lewis, 2004; Bozóki et al., 2016) , ecology and animal behavior (Laird & Schamp, 2006; Silk, 1999), and biology (Stuart-Fox et al., 2006; Sinervo & Lively, 1996). While the primary interest in these cases is typically in ranking agents/players, tournament graphs also reveal significant information about the nature of the game being played (Tuyls et al., 2018). This paper describes mathematical techniques for extracting useful information about the underlying game structure directly from tournament data. While these methods can be applied to the various contexts in which tournaments are already employed in machine learning (e.g., population based training), they open up a range of new research questions regarding the characterization of natural games, synthesis of artificial games (c.f. Omidshafiei et al. (2020)), game approximation via simplified dynamics, and the strategic perturbation of games.

Fine structural characteristics of a tournament graph can be represented by low dimensional embeddings that map competitive relationships to embedded geometry. We review and expand on methods introduced by Balduzzi et al. (2018b), who proposed a canonical series of maps that provide a complete description of a sample tournament in terms of a sum of simple games, namely, disc games. PTA provides a simplified global understanding of a tournament compatible with a broad set of objectives beyond finding equilibrium solutions.

Note that our objectives are as empirical as they are game theoretic. Empirical game theory, the study of games from actual game play data (e.g. sports analytics), studies games *as they are played* by a particular population, rather than by an idealised player. Thus, empirical game theory has its own, valid, objectives beyond finding equilibria or optimal players. Exclusive focus on optima ignores the global structure of a game as it is experienced by the majority of players. What decision dilemmas do they face? What game dynamics do they experience? What game space must they navigate in the process of optimization? How should they exploit a chosen opponent, population, or form teams? All of these questions are more easily addressed given a simplified global representation that isolates each important independent aspect of a game. PTA offers such a summary.

Our contribution follows. First, we compare PCA (Pearson, 1901) to disc game embedding, and show that disc game embeddings inherit key algebraic properties responsible for the success of PCA. Based on this analogy, we propose PTA as a general technique for visualizing data arising from competitive tasks or pairwise choice tasks. Indeed, while we focus on games for their charisma, any data set representing a skew-symmetric comparison of objects is amenable to PTA. Via a series of examples, we show that PTA provides a much richer framework for analyzing trade-offs in games than previously demonstrated. Our examples exhibit a wide variety of strategic trade-offs that can be clearly visualized with PTA. Unlike previous work, we focus on the relation between embedding coordinates, which represent performance relations, and underlying agent attributes in order to elucidate the principal trade-offs responsible for cyclic competition in each game. Moreover, we consider the full information content of PTA by analyzing multiple leading disc games and by studying the decay in their importance. Important strategic trade-offs can arise in later disc games, so previous empirical work's focus on the leading disc game is myopic. These examples also raise conceptual limitations not addressed in previous work, thus outline future directions for development.

## 2 RELATED WORK

Our work builds directly on Balduzzi et al. (2018b), which used the embedding approach to introduce a comprehensive agent evaluation scheme. Their scheme uses the real Schur form (PTA) in conjunction with the Hodge decomposition to overcome deficiencies in standard ranking models. Our work also compliments efforts to explore cyclic structures in competitive systems (Candogan et al., 2011; Strang et al., 2022b), economics (Linares, 2009; May, 1954), and tangentially as multiclass classification problems (Bilmes et al., 2001; Huang et al., 2006). Cycles challenge traditional gradient methods and can slow training (Omidshafiei et al., 2020; Balduzzi et al., 2018a). Moreover, cyclic structures in games are often intricate and difficult to disentangle, particularly among intermediate competitors. Games of skill frequently exhibit this "spinning top geometry" (Czarnecki et al., 2020). By summarizing cyclic structures, PTA helps identify areas of the strategy space that cause difficulty during training, or should be targeted for diverse team design (Balduzzi et al., 2019; Garnelo et al., 2021). Here, we show that PTA can identify fundamental trade-offs that summarize otherwise opaque cyclic structure. Trade-offs play an important role in decision tasks and evolutionary processes outside of games, so general tools that isolate and reify trade-offs are of generic utility (Omidshafiei et al., 2020; Tuyls et al., 2018). In that sense, our attempt to visualize game structure is in line with generic data visualization efforts, which aim to convert complicated data into elucidating graphics (c.f. Healy (2018); Garnelo et al. (2021)).

## 3 BACKGROUND

### 3.1 FUNCTIONAL FORM GAMES

A two-player zero-sum functional form game, is defined by an attribute space $\Omega \subseteq \mathbb{R}^T$ and an evaluation function $f$ that returns the advantage of one agent over another given their attributes. Agents in the game can be represented by their attribute vectors $x, y \in \Omega$, the entries of which could represent agent traits, strategic policies, weights in a neural net governing their actions, or more generally, any attributes that influence competitive behavior. The function $f$ is of the form $f : \Omega \times \Omega \to \mathbb{R}$. The value $f(x, y)$, quantifies the advantage of agent $x$ over $y$ with a real number. The evaluation function must be fair, that is, the advantage of one competitor over another should not depend on the order they are listed in. Consequently, $f$ must be skew symmetric, $f(x, y) = -f(y, x)$ (Strang et al., 2022b). If $f(x, y) > 0$ we say that $x$ beats $y$ and the outcome is a tie if

$f(x, y) = 0$. The larger $|f(x, y)|$ the larger the advantage one competitor possesses over another. We do not specify how advantage is measured, since the appropriate definition may depend on the setting. Possible examples include expected return in a zero-sum game, probability of win minus a half, or log odds of victory. With a set of agents $X$, pairwise comparisons of all agents gives a $N \times N$ evaluation matrix $F$ where $F_{ij} = f(x_i, x_j)$. Any such matrix can be separated into transitive and cyclic components, $F_t$ and $F_c$, via the Helmholtz-Hodge decomposition (HHD) (Balduzzi et al., 2018b; Strang et al., 2022b; Lim, 2020). The HHD writes $F = F_t + F_c$ where $[F_t]_{ij} = r_i - r_j$, and where $r$ are least squares ratings that evaluate the average performance of each agent. These matrices can be analyzed to study the structure of the game among those competitors, i.e. the resulting tournament.

## 3.2 DISC GAMES

The cyclic component of a tournament can be visualized using a combination of simple cyclic games (Balduzzi et al., 2019; 2018b). The simplest cyclic functional form game is a disc game, which acts as a continuous analog to rock-paper-scissors (RPS) in two-dimensional attribute spaces. The disc game evaluation function is the cross product between competitor's embedded attributes,

$$\text{disc}(x, y) = x \times y = x_1 y_2 - x_2 y_1 = x^T \begin{bmatrix} 0 & 1 \\ -1 & 0 \end{bmatrix} y = x^T R y \qquad (1)$$

where $R$ is the $2 \times 2$ ninety degree rotation matrix (Balduzzi et al., 2018b). The cross product models a basic trade-off between the two attributes. Any evaluation matrix can be represented with a sum of pointwise embeddings into a sequence of disc games. The necessary construction follows.

## 4 PRINCIPAL TRADE-OFF ANALYSIS (PTA)

PTA decomposes an arbitrary performance matrix $F$ into a sum of simpler performance matrices by embedding each agent into a series of disc games that model important strategic trade-offs.

Any real, $m \times m$, skew-symmetric matrix $A$ admits a Schur decomposition (real Schur form), $QUQ^T$. Here $Q$ is an orthonormal $m \times \text{rank}(A)$ matrix, $U$ is block diagonal with $\text{rank}(A)/2$, $2 \times 2$ blocks of the form $U^{(k)} = \omega_k R$, and where $\omega_k \geq 0$ is a nonnegative scalar. Each pair of consecutive columns of $Q$, $[q_{2k-1}, q_{2k}]$, correspond to the real and imaginary parts of an eigenvector of $A$ scaled by $\sqrt{2}$. The scalars $\omega$ are the nonnegative imaginary part of the corresponding eigenvalues, listed in decreasing order (Youla, 1961; Zumino, 1962). A linear algebra exercise demonstrates that the columns of $Q$ are also proportional to the singular vectors of $A$, and the sequence of scalars $\omega$ match the singular values of $A$. A truncated expansion of $A$ using only the first $r$ blocks is equivalent to truncating the singular value decomposition at $2r$ singular vectors, so equals the optimal rank $2r$ approximation to $A$ under the Frobenius norm via the Eckart-Young-Mirsky theorem (Eckart & Young, 1936; Mirsky, 1960; Strang, 2019).

When $A$ is replaced with the performance matrix $F$, each block in the Schur decomposition acts as a scaled version of a disc game where each competitor is assigned embedding coordinates via $Q$. The performance matrix $F$ us skew symmetric, so admits a Schur decomposition:

$$F = QUQ^T. \qquad (2)$$

As in PCA, we consider a low rank approximation of $F$ associated with expansion onto the first $k$ disc games, where $k$ is chosen large enough to satisfy a desired reconstruction accuracy. Low rank approximation allows model reduction and mixed equilibria approximation in quasi-polynomial time (Lipton et al., 2003). The closest rank $2r$ approximation to $F$ in Frobenius norm is given by replacing $Q$ with $Q^{(1:2r)}$, and $U$ with $U^{(1:2r)}$ in Equation 2, where $Q^{(1:2r)}$ is the first $2r$ columns of $Q$, and $U^{(1:2r)}$ is the upper $2r$ by $2r$ minor of $U$.

The matrix $Q^{(1:2r)}$ provides a set of basis vectors. Projection onto those basis vectors define a new set of coordinates, thereby embedding the competitors. Specifically, let:

$$\hat{Y} = Q^{(1:2r)^T} F = U^{(1:2r)} Q^{(1:2r)^T}. \qquad (3)$$

and scale each pair of embedding coordinates by the associated eigenvalue so $\vec{y}_k(i) = [y_{2k-1,i}, y_{2k,i}] = \omega_k^{-1/2}[\hat{y}_{2k-1,i}, \hat{y}_{2k,i}] = \omega_k^{1/2}[q_{2k-1,i}, q_{2k,i}]$. Then $\vec{y}_k(i)$ maps from competitor indices, $i$, to points in $\mathbb{R}^2$, and the set $Y = \{\vec{y}_k\}$ is a collection of planar embeddings, where $\vec{y}_k$ is given by projection onto a feature plane spanned by $q_{2k-1}$ and $q_{2k}$.

A user interested in the transitive and cyclic components of $F$ separately, could begin by breaking $F$ into $F_t$ and $F_c$ (Strang et al., 2022b). The transitive component can be represented on a line via the ratings, so does not require additional visualization (Balduzzi et al., 2018b). The cyclic component $F_c$ is skew symmetric, so can be represented via PTA. Then, performance is represented by a combination of two components. The first compares the overall quality of the agents, as quantified by a set of ratings. The second represents any cyclic relations as a combination of principal trade-offs. We apply PTA to $F$, not $F_c$ in all of our experiments.

Since PTA depends only on $F$, the cost of performing PTA is independent of the complexity of the underlying game or agents. Once $F$ is formed PTA proceeds at the same cost ($\mathcal{O}(n^3)$ for $n$ sampled agents Golub & Van Loan (2013)) as standard low-rank matrix decompositions, like PCA, which seek the leading singular vectors of a symmetric matrix. While iterative algorithms may be more efficient when only a few leading columns of $Q$ are required, the computational cost of performing PTA in an empirical setting will almost always be swamped by the cost of gathering the data for forming $F$, which requires evaluating $\mathcal{O}(n^2)$ pairs of agents. The cost of performing PTA is inconsequential since any computational cost comes from all simulations needed to compute F. The simulation cost could be reduced if low-rank completion methods were applied to fill in missing data. We leave sampling considerations and matrix completion methods (Meka et al., 2009; Gleich & Lim, 2011; Chen & Joachims, 2016) to future work.

Note that the Schur decomposition is only unique up to rotation within each feature plane, since complex conjugate pairs of eigenvectors of $F$ are only uniquely defined up to their complex phase. Thus, two embeddings are equivalent if they agree up to rotation within each planar embedding.

The evaluation $F_{ij}^{(2k)}$ between agent $i$ and agent $j$ equals a sum over each embedding $\vec{y}_k$, of the cross product $\vec{y}_k(i) \times \vec{y}_k(j)$ (see Appendix B). That is:

$$F_{ij}^{(2r)} = \sum_{k=1}^{r} \vec{y}_k(i) \times \vec{y}_k(j) = \sum_{k=1}^{r} \text{disc}(\vec{y}_k(i), \vec{y}_k(j)). \tag{4}$$

Thus, restricted to each planar embedding $F^{(2r)}$ is a disc game and the optimal rank $2r$ approximation of $F^{(2r)}$ is a linear combination of disc games applied to the sequence of planar embeddings $\{\vec{y}_k\}_{k=1}^{r}$.

This decomposition is useful for two reasons. First, it depends on a spectral decomposition of $F$, so inherits the key properties that account for the success of PCA. An equivalent construction is introduced in Chen & Joachims (2016) where it is called the "blade-chest-inner" model. The construction in Chen & Joachims (2016) is not based on a spectral decomposition, so lacks orthogonality or low rank optimality.

In PTA, the embeddings are projections onto orthogonal planes, so each embedding encodes independent information about cyclic competition. Here independence means that the embedding coordinates of a randomly sampled agent in two distinct planes are uncorrelated. Note that this definition depends on the density of sampled agents. The planes act like feature vectors, and are typically associated with some strategic trade-off (see Section 5). Therefore, as PCA identifies principal components, PTA identifies principal trade-offs: orthogonal planes associated with a sequence of fundamental cyclic modes. The two decompositions differ since PCA uses the singular value decomposition, while PTA uses the Schur real form. Nevertheless, the sequence of embeddings form optimal low rank approximations to $F$, where the importance of each embedding is quantified by the magnitude of the associated eigenvalue. Thus, the sequence of eigenvalues determines the number of disc game embeddings, $r$, required to achieve a sufficiently accurate approximation of $F$. The number of disc games is half the effective rank of $F$, and is a natural measure of the complexity of cyclic competition. The complexity is distinct from the overall intensity of cyclic competition or the game balance, which depend on $\|F\|$ instead of its rank (Strang et al., 2022b). Instead, it counts the number of distinct cyclic modes in the evaluation matrix. It is possible to have many distinct, yet weak cyclic trade-offs, or one, very strong, cyclic trade-off.

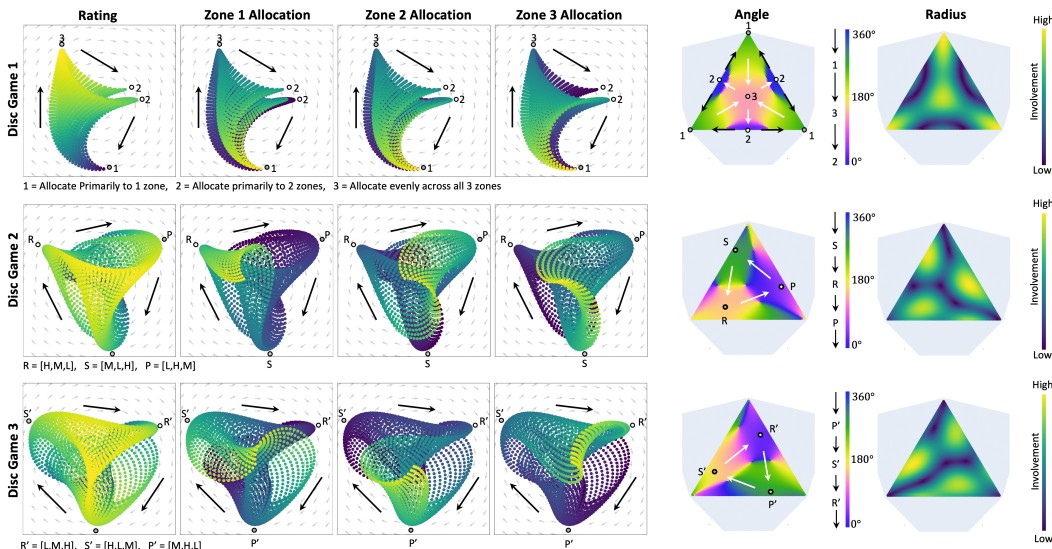

Figure 1: Disc games 1 to 3 of Blotto [1,1,1] game with $N = 75$. **Rows:** disc game number. **Columns 1 - 4:** disc game embeddings colored according to agent rating, then agent allocation to zones 1 to 3. **Column 5:** the angle (measured counterclockwise from the horizontal axis), of the embedding of each strategy. Advantage flows clockwise in angle, so blue beats purple beats yellow beats green beats blue. **Column 6:** the radius of the embedding of each strategy. High radius corresponds to strong involvement in a trade-off (yellow), while small radius corresponds to low involvement (blue). Each triangle in the fifth and sixth columns represents the space of available allocations. High allocations to zone 1 cluster near the bottom left corner, high allocations to zone 2 cluster near the bottom right corner, and high allocations to zone three cluster at the top corner. **Labels:** Representative allocations defining the underlying trade-offs, indicated with bold arrows (see Table 2).

Second, the disc game encodes performance relations via geometry. Given coordinates $\vec{y}(i)$ and $\vec{y}(j)$, the advantage of competitor $i$ over competitor $j$ given by $\vec{y}(i) \times \vec{y}(j)$ equals twice the signed area of the triangle with vertices at the origin, $\vec{y}(i)$, and $\vec{y}(j)$. In polar coordinates, each agent is assigned a radius and angle in each disc game, $(r_k(i), \theta_k(i))$. Then, $\text{disc}(\vec{y}(i), \vec{y}(j)) = r_k(i)r_k(j)\sin(\theta_k(j) - \theta_k(i))$. So, the larger $r_k(i)$, the more the $k^{th}$ cyclic mode influences the performance of agent $i$. For a fixed radius, one competitor gains the most advantage when it is embedded $90°$ clockwise from its opponent, and possesses an advantage as long as it is embedded clockwise of its opponent. Thus, advantage flows clockwise about the origin. We visualize this flow with a circulating vector field, $\vec{v}(\vec{y}) = [y_2, -y_1]$. These geometric properties allow disc games to encode a variety of cyclic structures in interpretable visuals.

## 5 EXPERIMENTS

Here, we illustrate the graphical power of PTA via Blotto and Pokemon. Both exhibit interesting cyclic structure. We emphasize the interpretation of each principal trade-off in terms of game strategy to show that PTA reveals diverse, fine-grained game structure based only on empirical game data. A simpler example is provided in Appendix D for reference.

### 5.1 COLONEL BLOTTO

Colonel Blotto is a zero-sum, simultaneous action, two player resource allocation game (Kovenock & Roberson, 2021). Each player possesses $N$ troops to distribute across $K$ zones. Each zone has an associated payout $Z_k$. A zone is conquered by a player if they allocate more troops to that zone than their opponent. The conquering player receives the payout. Ties result in both players receiving

Table 1: Principal Trade-Offs

| D.G. | Allocation Types | Location in Simplex | Example | Advantage Relation |
|---|---|---|---|---|
| 1 | $(1)$ = allocate to 1 zone
$(2)$ = allocate to 2 zones
$(3)$ = allocate to 3 zones | corners
midpoints of edges
center | $[\mathbf{70}, 0, 5]$
$[\mathbf{38}, \mathbf{37}, 0]$
$[\mathbf{25}, \mathbf{25}, \mathbf{25}]$ | $(1) < (3) < (2) < (1)$ |
| 2 | $R = [H, M, L]$
$S = [M, L, H]$
$P = [L, H, M]$ | corners
shifted
counter clockwise | $[\mathbf{50}, 25, 0]$
$[25, 0, \mathbf{50}]$
$[0, \mathbf{50}, 25]$ | $R < P < S < R$ |
| 3 | $R' = [L, M, H]$
$S' = [H, L, M]$
$P' = [M, H, L]$ | corners
shifted
clockwise | $[0, 25, \mathbf{50}]$
$[\mathbf{50}, 0, 25]$
$[25, \mathbf{50}, 0]$ | $R' < P' < S' < R'$ |

Principal trade-offs associated with the first three disc games. The columns list the allocation types involved in each disc game (D.G.), their location in the simplex of possible allocations, provide an example set of allocations, and the competitive relations between the types. The letters H, M, and L, are used to denote high, medium, and low allocation. Note that the allocations involved in the trade-off defined by a disc game correspond to locations in the radius panels in Figure 1 shaded green or yellow. Advantage in a disc game flows clockwise, so can be inferred from the angle panel.

0 payout. The player with the highest total payout wins the match. All allocations are revealed simultaneously.

At simplest, the payouts are uniform across zones, so the player who conquers the most zones wins the game. Unweighted Blotto is a highly cyclic game since there is no dominating strategy. Every strategy admits a counter. Unless $K = N$ or $K <= 2$, all allotments lose to some other allotment. To defeat an allotment, adopt the maxim, "lose big, win small". Mimic the allotment, then redistribute all the units from the zone with the most units as uniformly as possible across the remaining zones. Then, unless all zones were allotted one unit, the exploiting strategy sacrifices a loss in one zone to win in more than one other zone. In general, the more an allotment commits to a single zone, the more easily it is defeated. Unweighted Blotto is also complex, since the zones are indistinguishable. Thus unweighted Blotto admits $K!$ fold symmetries with respect to the zone labels.

We consider each unique strategy as a separate "agent", parameterized by the corresponding allotment. We generate agents by randomly sampling over the strategy space using a Dirichlet distribution with the support equal to the number of zones. After sampling, we compare each pair of strategies in the population. Each match-up is deterministic and results in a win, loss or tie, which we assign scores (0.5,0,-0.5). We construct the associated evaluation matrix by setting $F_{ij}$ to the score of strategy $i$ against strategy $j$.

PTA allows elegant visualization of relevant game structure by reducing a game to a small set of key trade-offs. We start by looking at the $K = 3$, $N = 75$ blotto game with uniform payouts. Table 2 summarizes the principal trade-offs associated with each disc game. These trade-offs are the most important sources of cycles in the tournament, accounting for 80% of its structure.

In general, the number of distinct allotments in a $K$ battlefield, $N$ troop blotto game grows at $\mathcal{O}(N^K)$, but the complexity, which reflects the underlying number of cyclic modes, converges to a constant value associated with a continuous Blotto game, where commanders can allocate an arbitrary fraction of their force to each zone. Unweighted $K = 3$, $N = 75$ blotto admits 2926 allotments, but has a 3! fold exchange symmetry under permutations of the battlefield labels, leaving roughly 488 distinct allotments. Three disc games reconstruct the evaluation matrix to $\approx 80\%$ accuracy, 6 to $\approx 90\%$ accuracy, and 12 to $\geq 95\%$ accuracy, so the game has complexity 12 at a 95% standard. Trade-offs 4 - 12 represent refinements of the trade-offs present in the first three disc games, so PTA really allows a reduction in complexity from 2926 allocations (absent prior knowledge regarding symmetries), to 3 fundamental cyclic modes. Thus, PTA can effectively separate the underlying complexity of a game from the size of its strategy space. See Appendix E for further discussion.

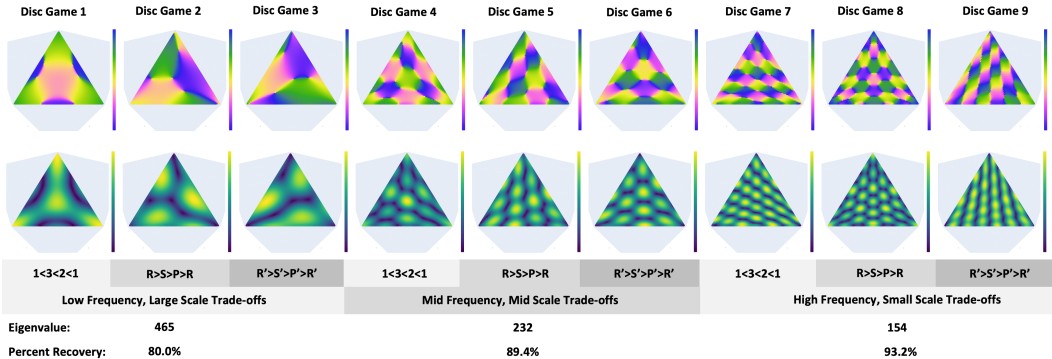

Figure 2: The first nine disc games of the $N = 75$, [1,1,1] blotto game. Each column is a separate disc game. The first and second rows show the angle and radius assigned to each allocation. The disc games are labelled by trade-off type. Consecutive sets of three share the same eigenvalue and are grouped by spatial scale with eigenvalue and percent recovery of $F$ provided beneath.

The exchange symmetry of the zones is apparent in the sequence of eigenvalues, $\omega_k$, representing disc game importance. Exchanges introduce 6 permutations under which the evaluation matrix is invariant. Consequently, $\omega_k$ come in sets of three, where each $\omega_k$ represents a pair of eigenvectors. Eigenvectors associated with identical eigenvalues are not uniquely defined. Instead, they are drawn from a subspace of dimension equal to the multiplicity of the repeated eigenvalue. Consequently, all of the eigenvectors $Q$ are chosen arbitrarily from six dimensional spaces.

When $F$ has repeated eigenvalues, the associated disc game embeddings are not unique. Any unitary transform of the set of eigenvectors sharing an eigenvalue defines a valid embedding. Thus, symmetry presents an unusual challenge: degeneracy. In our case, the disc games come in sets of three, each representing an arbitrary rotation of a six dimensional object. Consequently, we consider multiple disc games simultaneously. This issue was not addressed in previous work, which largely only considered the leading disc game. Generic games should not exhibit strong symmetries, so such degeneracy will be rare and confined to toy examples. That said, generic games also require more than one disc game, so it is essential to consider more than the leading disc game.

We analyze the three leading disc games to identify the most important allocation trade-offs. Figure 1 shows the first three disc games colored by rating, allocation to the three zones, and the mapping to angle and radius in each disc game as a function of allocation. Each share the same eigenvalue, so are equally important and could be mixed. Nevertheless, these three disc games represent distinct trade-offs in allocations that can be easily explained.

The specific trade-offs can be identified directly from the disc games when colored by allocation. Consider the points labelled 1, 2, and 3 in the first disc game. Each maximize the radius of the scatter cloud while moving along its boundary, so represent the allocations most involved in the cycle. The low rated points at the bottom of the scatter allocate primarily to one zone (yellow in panels 2 - 4).

Moving clockwise, the next extrema occurs at the top of the scatter. It is high rated, and has nearly equal allocation across all three zones (colored green in panels 2 - 4). Uniform allocations are rated highly since they perform well against most randomly sampled allocations, particularly those lying along a line connecting a corner of the simplex to its center. This induces a transitive trend among the bulk of the allocations moving from allocations that prioritize one zone, to allocations that treat the zones equally. This transitive trend is represented by the general shift of the disc game leftward off the origin. This subset of allocations compete transitively, producing the regular gradient from purple to yellow in rating when moving clockwise from the bottom to the top in the scatter.

Not all allocations satisfy this transitive trend. Allocations that prioritize two zones counter the uniform strategy, and are countered by allocations that prioritize a single zone. For example, allocation [70,0,5] defeats [38,37,0]. Thus, allocations lying on the midpoints of an edge of the simplex lose to allocations near either neighboring endpoint. These counters close the cycle, and are represented

Figure 3: Disc games 1,2 and 4 for pokemon. Disc game one is colored by rating. Disc game two is colored by type, then generation. Disc game four is colored by rating, then generation.

by the rightmost pair of corners labelled 2 in disc game 1. Panels 2 - 4 show that each such corner receives an intermediate allocation in two zones (green), but little to none in the third (dark blue).

Similar visual analysis identifies the RPS cycles among cyclic permutations of allocations [H,M,L] and [L,M,H] shown in disc games 2 and 3. For example, the leftmost corner of the scatter cloud shown in disc game 2 receives a high allocation in zone 1 (teal), an intermediate allocation in zone 2 (blue-green), and a low allocation in zone 3 (dark blue). Walking from R to P to S, the allocation patterns shifts cyclically. The same analysis applies to disc game 3, starting from [L,M,H].

Figure 2 shows the angle and radius assigned to each allocation in the simplex. Strikingly, subsequent disc games imitate the disc game 1-3 trade-offs, only at higher frequency on a smaller spatial scale in allocation. This suggests that the disc games may act like Fourier modes, where early disc games capture low frequency, global trade-offs, and later disc games capture high frequency, local trade-offs. It also suggests that orthogonality may not be the appropriate notion of independence for trade-offs. A sharper notion of equivalency is needed. Methods like nonnegative matrix factorization, which address similar issues among PCA features (Lee & Seung, 1999), suggest an avenue for further development. An example that produces explicit sine series is discussed in the Appendix.

## 5.2 POKEMON

We conclude by analyzing Pokemon. Pokemon originated from the Nintendo Game Boy console, but has since been played on a variety of mediums including playing cards. Pokemon is of considerable interest from a game design perspective since the creators must design certain trade-offs to keep the game balanced and engaging. The game is made up of creatures, called Pokemon, that come in many varieties. Players are rewarded for collecting diverse teams. Thus, each Pokemon has a different type, and each type has its own set of strengths and weaknesses. These different types satisfy interlocking cyclic relationships.

The data used in this analysis comes from an open-source Kaggle data set (Bouchet, 2017). The original data has 800 Pokemon, but we removed the 65 "legendary" Pokemon to simplify the analysis. The data consists of battle outcomes and pokemon attributes. Battle outcomes were converted into an evaluation matrix by logistic regression (see Appendix F).

Figure 3 shows three of the first four disc games, chosen for their significance. The first disc game is the most important, and is clearly transitive since all points fall on a curve that does not enclose the origin. Position along the curve is closely correlated with speed, so speed determines rating.

We query by attribute to interpret the remaining disc games. To start, consider the "type" attribute. The second disc game is clearly clustered by type (see Figure 3). A variety of RPS relationships are apparent among the type clusters. Any loop of clusters containing the origin corresponds to a cycle of type advantage. The intensity of the corresponding cycle (curl) is proportional to its area. Focus on the large clusters most involved in the trade-off, i.e. furthest from the origin. Figure 4 summarizes the RPS relations between these clusters. First, notice the highlighted triangle formed by the Water-Fire-Grass clusters. The disc game shows the expected advantage cycle since the triangle contains the origin. Thus, PTA identifies known game structures without domain specific knowledge.

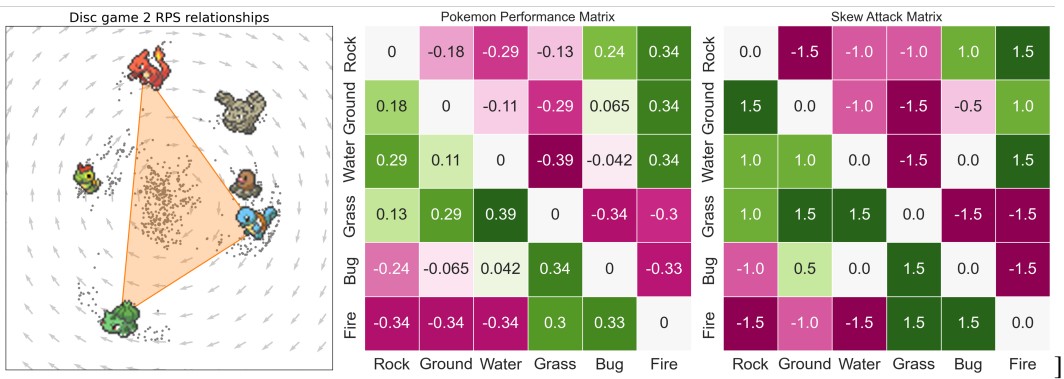

Figure 4: **Left:** RPS sub-game discovery. Each cluster type is represented by a matching pokemon, **Middle:** Empirical performance matrix, **Right:** Performance matrix derived from type chart

Additional clusters on the outer ring satisfy more intricate relations. The other three types are "bug", "rock" and "ground". To summarize these relations we construct a coarse grained evaluation matrix, $\hat{F}$. Specifically $\hat{F}_{ij}$ is the average performance of Pokemon from type $i$ vs the Pokemon from type $j$ in the second disc game. The associated matrix heat-map is shown in the middle panel of Figure 4. The types are ordered by angle moving clockwise about the origin.

We compared these relations with available game design matrices known as "attack matrices". Attack matrix entries list the advantage of one Pokemon type over the other. We use the attack matrix from (type chart). An attack matrix is written in terms of multiples, so Pokemon that are evenly matched have a $1\times$ advantage. We bucket the range of $i, j$ attack multipliers $a_{ij}$ into 5 bins ranging from $0\times$ to $2\times$, skew-symmetrize via $(A - A^T)$. The result is the rightmost panel in Figure 4.

The coarse grained summary $\hat{F}$ is strikingly similar to the provided attack matrix. The apparent structural parity in these two matrices highlights the virtues of PTA. Without any domain knowledge, access to attributes, or any explicit instruction to identify clusters, PTA clustered Pokemon by their most relevant attributes (type) then encoded a game mechanism (type specific attack multipliers) directly from the cluster locations. Conversely, the second disc game shows how cyclic relations introduced at the mechanism level are realized in actual performance.

Coloring the disc games by "generation", i.e. pokemon release date, reveals design choices. The game is frequently updated by the addition of new Pokemon. Updates present a design challenge. Game designers must introduce desirable new Pokemon without upsetting the game balance. The fourth disc game, shown in the far right plot of Figure 3, is balanced in that rating does not predict angle, and instead correlates with radius. Strong and weak Pokemon are closer to the origin, while Pokemon of intermediate rating are more involved in the trade-off. This reveals a spinning top structure characteristic of many games (Czarnecki et al., 2020). Rather, generation predicts angle. Each generation possesses an advantage over its predecessor, as illustrated by the fade from purple to yellow. Balance is retained since generational advantage is periodic. The same clockwise generation shift reappears in the second disc game. Within type, new beats old. For example, the bottom-most cluster (grass) clearly trends old to young. Cross type relations are largely unchanged.

## 6 CONCLUSION

Following Balduzzi Balduzzi et al. (2018b), we have demonstrated that all evaluation matrices admit an expansion onto a sum of disc game embeddings. We suggest the name PTA based on the close analogy with PCA. Through examples, we have demonstrated that embeddings produced by PTA can reveal a surprising variety of competitive structures from outcome data alone. Future work could provide more general methods for finding embeddings, such as a functional theory connecting performance with attribute space, or could seek a sparser representation via extensions of sparse PCA (Zou et al., 2006). Future work should also investigate automated methods that summarize the trade-offs identified by PTA without the need for visual inspection, and that leverage the representation for game classification, construction, and exploration.

## 7 REPRODUCIBILITY

In the supplementary material we have included the necessary code to be able to reproduce the experiments from the main section of the paper. Included is a README file with instructions on how to run the code.

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

## A  APPENDIX (SUPPLEMENTARY MATERIAL)

## B  PRINCIPAL TRADE-OFF ANALYSIS

### B.1  SCHUR DECOMPOSITION IS A SUM OF DISC GAMES

Here we prove that the Schur decomposition (real Schur form), is equivalent to a sum of disc games applied to the embedding maps $\vec{y}_k$.

Recall the embedding construction. Given a skew symmetric matrix $F$, write $F = QUQ^T$ where $Q$ is real, orthonormal, $U$ is block diagonal with diagonal blocks $\omega_k R$, and $R$ is the two by two ninety degree rotation matrix. Let $\vec{y}_k(i) = \omega_k^{1/2}[q_{i,2k-1}, q_{i,2k}]$. Then, the rank $2r$ approximation to $F$ is:

$$
\begin{aligned}
F_{ij}^{(2r)} &= e_i^T \left( \sum_{k=1}^{r} w_k [q_{2k-1}; q_{2k}]^T R [q_{2k-1}; q_{2k}] \right) e_j \\
&= \sum_{k=1}^{r} w_k [q_{i,2k-1}; q_{i,2k}]^T R [q_{j,2k-1}, q_{j,2k}] \\
&= \sum_{k=1}^{r} w_k (q_{i,2k-1} q_{j,2k} - q_{i,2k}, q_{j,2k-1}) \\
&= \sum_{k=1}^{r} \sqrt{w_k}[q_{i,2k-1}, q_{i,2k}] \times \sqrt{w_k}[q_{j,2k-1}, q_{j,2k}]
\end{aligned}
\tag{5}
$$

Recalling the embedding construction, write:

$$
F_{ij}^{(2r)} = \sum_{k=1}^{r} \vec{y}_k(i) \times \vec{y}_k(j) = \sum_{k=1}^{r} \mathrm{disc}(\vec{y}_k(i), \vec{y}_k(j)).
\tag{6}
$$

Thus, restricted to each planar embedding $F_c^{(2r)}$ is a disc game and the optimal rank $2r$ approximation of $F^{(2r)}$ is a linear combination of disc games applied to the sequence of planar embeddings $\{\vec{y}_k\}_{k=1}^{r}$. $\square$

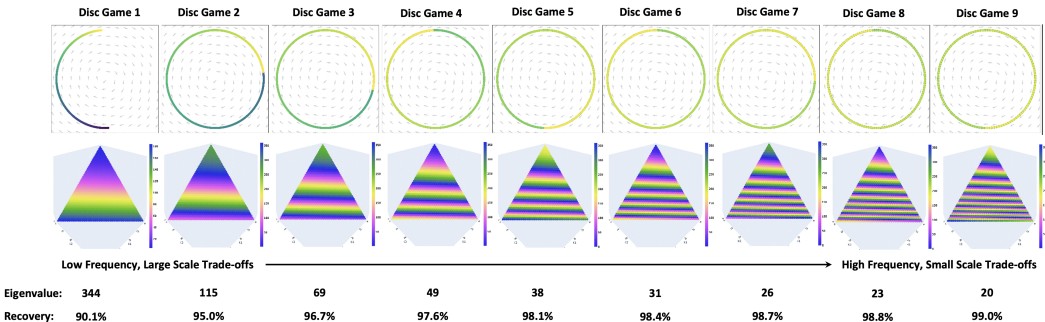

Figure 5: **Top row:** The first 9 disc games for $[1, 2, 4]$ blotto, colored by rating. **Bottom row:** Phase as a function of allocation in each disc game. The top corner of the triangle corresponds to exclusive allocation to zone 3. Note that phase allocation increases in frequency with increasing disc game. Colors denote phase, with advantage flowing clockwise. The eigenvalues associated with each disc game are provided, along with the percent of $F$ recovered by the running sum of the disc games. Note that the first disc game is responsible for 90% of the structure of $F$, and the first two disc games account for 95% of its structure.

### B.2 PTA AND FOURIER SERIES

Both the $[1, 1, 1]$ and the $[2, 3, 4]$ blotto examples exhibit strikingly modal disc games that repeat at increasing frequency, and on smaller spatial scales, with increasing disc game number. These patterns suggest an analogy to Fourier series. To make the analogy more concrete we present one last example.

Consider $[1, 2, 4]$ blotto. Since the net value of the first two zones is less than the value of the fourth zone, a player wins the overall game if they they win the third zone, or, tie in the third zone and win the second. Otherwise they tie in all zones or lose. Thus, the performance function $f([x_1, x_2, x_3], [y_1, y_2, y_3]) = \text{sign}(x_3 - y_3) + \chi_{z_3=0}(x - y)\text{sign}(x_2 - y_2)$ where $\chi(z)$ is the indicator function for the event in the subscript. Performance can be reduced to a comparison of a single agglomerated trait, $w(x) = x_3 + \frac{1}{N-x_3+1}x_2$. Then $f(x, y) = \text{sign}(w(x) - w(y))$. Thus, performance is a step function applied to the difference $w(x) - w(y)$. The difference is dominated by the difference in allocations to the third zone.

Figure 5 shows the first nine disc games. Notice that all allocations are embedded onto circles, or, in the first case, half circles. Moreover, phase (position along each circle), is entirely a function of the agglomerated trait $w(x) = x_3 + \frac{1}{N-x_3+1}x_2$. This form is apparent in the phase panels, where phase is close to constant for fixed allocation to zone 3, but is tilted slightly to account for allocation to zone 2. The first disc game is transitive and completes one half circle moving clockwise from zero allocation to zone 3 to exclusive allocation to zone 3. Discs games 2 and 3 complete 1.5 and 2.5 rotations each when moving from zero allocation to zone 3 to exclusive allocation to zone 4. The pattern continues for the first 9 disc games. Moreover, the circle radii decay geometrically (as shown by the sequence of gold scatter points in the last panel of Figure 8).

These features are hallmarks of a sine series embedding. We show below that any translationally invariant performance function of a single attribute can be represented by a sum of disc games, where the embedding into each disc game maps to a circle, the attribute maps to a phase coordinate around the circle, and the radii of the circles in each disc game are controlled by the coefficients of a sine series expansion. The performance function $f(x, y)$ only depends on the difference in allocations, $x - y$, so is translationally invariant, and is a function of a single trait, $w(x)$. Thus, it admits a sine series expansion. Note, the subsequent analysis does not guarantee low rank optimality, so only shows that disc game embedding via sine series is possible, not that PTA will necessarily produce such an embedding.

Consider a performance function of the form:

$$f(x, y) = A \sin(2\pi\omega(x - y)) \tag{7}$$

for $x, y \in \Omega \subset \mathbb{R}$, for arbitrary amplitude $A$, and frequency $\omega$.

Performance functions of this form are easy to embed, since the disc game uses a cross product. The cross product between two points on a plane, expressed in polar coordinates, is the product of their radii times the sine of the difference in their phases. Therefore, if:

$$\vec{y}(x) = \sqrt{|A|}[\cos(2\pi \text{sign}(A)\omega x), \sin(2\pi \text{sign}(A)\omega x)] \tag{8}$$

then:

$$\begin{aligned} \text{disc}(\vec{y}(x), \vec{y}(y)) &= \sqrt{|A|}^2 \sin(2\pi \text{sign}(A)\omega(x-y)) \\ &= \text{sign}(A)|A| \sin(2\pi \omega(x-y)) = A \sin(2\pi \omega(x-y)) = f(x,y). \end{aligned} \tag{9}$$

**Lemma 1: [Trigonometric Performance Functions of One Trait]** *If $f(x,y) = A \sin(2\pi \omega(x-y))$ for $x, y$ both in a one-dimensional trait space, then $f$ is disc game embeddable using the embedding:*

$$\vec{y}_k(x) = \sqrt{|A|}[\cos(2\pi \text{sign}(A)\omega x), \sin(2\pi \text{sign}(A)\omega x)]$$

.

Notice that this construction maps the intervals of length $1/\omega$ in $\Omega$ to the circle of radius $\sqrt{|A|}$ centered at the origin. It follows that, if a performance function is embeddable onto a circle centered at the origin then there exists a mapping from trait space to the real line where performance is of the form 7.

This result extends easily to linear combinations of sinusoidal functions with varying frequencies. Consider a performance function of the form:

$$f(x,y) = \sum_{k=1}^{n} A_k \sin(2\pi \omega_k (x-y)) \tag{10}$$

Then, $f$ can be recovered using a sum of $n$ disc game embeddings, where the $k^{th}$ embedding has the form:

$$\vec{y}_k(x) = \sqrt{|A_k|}[\cos(2\pi \text{sign}(A_k)\omega_k x), \sin(2\pi \text{sign}(A_k)\omega_k x)] \tag{11}$$

Note that, all the performance functions of this kind are translation invariant since they are functions of the difference $x - y$, which does not change if after shifting $x$ and $y$ by some amount $s$. :

**Theorem 1: [Translation Invariant One Trait Performance Functions]** *Suppose that $\Omega$ is a one-dimensional trait space, and $f(x,y)$ is translation invariant. Then there exists a function $h$ such that $f(x,y) = h(x-y)$. Suppose in addition that $h(x)$ is periodic with period $P$, or $\Omega$ is contained inside an interval with length $P/2$. Then $f$ is disc game embeddable using a countably infinite sequence of disc games, which correspond to the sine series expansion of $h$ and converge under the same conditions as the sine series. Moreover, each disc game represents a term in the sine series, and maps $\Omega$ to a subset of a circle centered at the origin with radius fixed by the corresponding coefficient in the sine series.*

**Proof:** If $f(x,y)$ is translation invariant then $f(x,y) = h(x-y)$ for some function $h$. Since $f(x,y) = -f(y,x)$, $h$ must be an odd function. If $\Omega$ is contained inside an interval of length $P/2$, then $h$ can be extended to an odd, continuous, $2P$ periodic function, or an odd $P$ periodic function. If not, then, by assumption, $h$ is periodic with period $P$.

All integrable $P$ periodic functions on the real line can be approximated with a Fourier series. If the function is real valued and odd, then the Fourier series is a sine series of the form:

$$h(x) \simeq \sum_{k=1}^{\infty} A_k \sin(2\pi \omega_k x), \quad \omega_k = \frac{k}{P}. \tag{12}$$

Each term in the sine series can be reproduced by a disc game embedding using the method for embedding sinusoidal functions introduced before. Specifically, let:

$$\vec{y}_k(x) = \sqrt{|A_k|}[\cos(2\pi \text{sign}(A_k)\omega_k x), \sin(2\pi \text{sign}(A_k)\omega_k x)] \tag{13}$$

where $A_k$ is the $k^{th}$ amplitude in the sine series of $h$:

$$A_k = \frac{4}{P} \int_0^{P/2} h(x) \sin(2\pi \omega_k x) dx. \tag{14}$$

Then, a partial expansion in terms of $r$ disc games equals the $r$ term sine series expansion of $h(x-y)$:

$$\sum_{k=1}^{r} \text{disc}(\vec{y}_k(x), \vec{y}_k(y)) = \sum_{k=1}^{n} A_k \sin(2\pi\omega_k(x - y)). \tag{15}$$

Thus, convergence of the sequence of disc game embeddings follows convergence of the sine series expansion. $\square$

It remains to show that, after sampling a finite set of agents, the result of PTA recovers the sine series representation. Sine series are low rank optimal in this case since, if the agents are ordered by increasing $w(x)$, the evaluation matrix is of the form:

$$F = \begin{bmatrix} 0 & 1 & 1 & \dots & 1 \\ -1 & 0 & 1 & \dots & 1 \\ -1 & -1 & 0 & \dots & 1 \\ \vdots & \vdots & \vdots & \ddots & \vdots \\ -1 & -1 & -1 & \dots & 0 \end{bmatrix} \tag{16}$$

This matrix is real, skew-symmetric, Toeplitz, and is diagonalized by the discrete Fourier transform, so PTA produces a sine series. The sequence of disc games act as the sine series expansion of a step function, with each higher order disc game corresponding to a higher order correction of an approximation to the step function. The first disc game is transitive and captures 90% of the structure of the evaluation matrix. Subsequent disc games correct the first disc game in order to produce a step function. While it only takes two disc games to recover 95% of the structure of $F$, so the 95% complexity of $[1, 2, 4]$ blotto is 2, the subsequent eigenvalues decay slowly, so stricter accuracy requirements lead to large complexities. The slow decay of the eigenvalues is a natural consequence of the slow convergence of sine series to a step function. Here it is clear that the complexity predicted by PTA overstates the complexity of the underlying game, since subsequent disc games are best interpreted as corrections that gradually finesse the first disc game, not distinct trade-offs.

More general proof and exploration is saved for future work.

## C  DISC GAMES

### C.1  GEOMETRY

Principal trade-off analysis is a useful visualization technique since disc games encode performance relations via embedding geometry. Reading a disc game requires familiarity with this geometry. Namely, familiarity with the various interpretations of a cross product. Here we review some relevant relations.

Cross products are closely related to area in the embedding. Given a pair of competitors with embedding coordinates $\vec{y}_k(i)$ and $\vec{y}_k(j)$, the performance of $i$ against $j$ in disc game $k$, $\text{disc}(\vec{y}_k(i), \vec{y}_k(j))$, equals the signed area of the triangle with vertices $\vec{y}_k(i)$, $\vec{y}_k(j)$, and the origin. Further, the degree of cyclicity on a loop $\mathcal{C}$ of competitors can be computed by evaluating a path sum of the advantages around the loop, i.e. curl($\mathcal{C}$) (Strang et al., 2022b). The curl on loop $\mathcal{C}$ equals the signed area of the loop traced out in each embedding (Strang et al., 2022a), summed over the embeddings. It follows that curl inherits the invariances of areas. In particular, curl is translation invariant.

The cyclic component $F_c$ on a given edge $i, j$ equals the average curl over all possible triangles formed by drawing a random $k$. Since curl is translation invariant, the cyclic component of competition is translation invariant. In contrast, the transitive component of competition is not translation invariant, and translation in a disc game induces a transitive component of competition (Balduzzi et al., 2018b). By subtracting $F_t$ from $F$ to recover $F_c$ we center all the rows and columns, so the scatter cloud of embedded competitors will be centered at the origin. In contrast, if we embed $F$ directly, then transitivity arises from translation of each scatter cloud away from the origin. If the origin is not included in the convex hull of the embedded agents, then competition is transitive.

In contrast, scaling the embedding coordinates does change the predicted performance relations. Area is proportional to length squared, so scaling the embedding coordinates by $\sqrt{s}$ scales the asso-

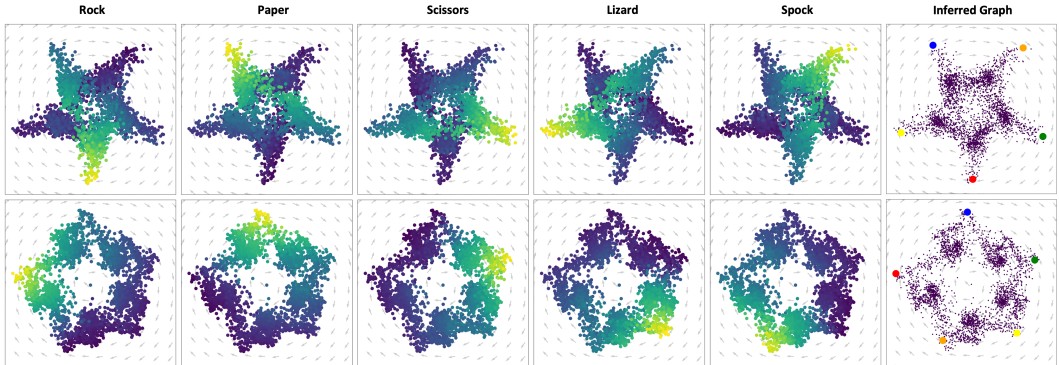

Figure 6: The two disc games that represent RPS+2 are shown. Starting from the first column the points are colored by the label for that column. The graph on the left shows a reduced representation of the game by using representative points for each strategy (red for rock, blue for paper, green for scissors, yellow for lizard, orange for spock).

ciated cyclic component of competition by $s$. The scaling from $\hat{Y}$ to $Y$, was adopted to ensure that unit area in embedding generates unit curl. It follows that the area encompassed by a set of points in a disc game embedding directly represents the amount of cyclic competition among those agents, and hence the importance of that embedding.

## D    ROCK PAPER SCISSORS + 2

Here we provide an extended example of the popular rock-paper-scissors (RPS) game. We consider rock-paper-scissor-lizard-spock (RPS+2). The utility matrix for (RPS+2) is shown below:

$$
U_{RPS+2} = \begin{bmatrix} 0 & -1 & 1 & -1 & 1 \\ 1 & 0 & -1 & 1 & -1 \\ -1 & 1 & 0 & -1 & 1 \\ 1 & -1 & 1 & 0 & -1 \\ -1 & 1 & -1 & 1 & 0 \end{bmatrix} \tag{17}
$$

We generate a population of agents using fictitious self play (FSP). Each agent in the population is defined by a vector of length 5 representing a distribution over the strategy space. We start with an initial random agent and generate best response agents using FSP. Each best response agent becomes a new agent in the population. We then create $F$ by computing the expected value of each match-up using the utility matrix equation 17.

What should we expect to see? All mixed strategies are an interpolation of the 5 pure strategies so each mixed strategy should be contained inside the convex hull of a polygon formed by embedding the pure strategies. The utility matrix is invariant under cyclic permutations, so the polygon must be regular and centered at the origin. Thus, the pure strategies must be the vertices of a regular pentagon. The full game includes two, equally important, interlocking cyclic relations among the pure strategies. These are, rock $\succ$ paper $\succ$ scissors $\succ$ lizard $\succ$ spock $\succ$ rock, and rock $\succ$ lizard $\succ$ paper $\succ$ spock $\succ$ scissors $\succ$ rock, where $\succ$ denotes loses to. Both cycles contribute equally to the utility matrix, and the utility matrix is unchanged under a permutation that rearranges the strategy labels so that rock is follow by lizard, then paper, then spock, then scissors.

These two cycles are represented by a pair of disc games, shown in Figure 6. In both, the pure strategies are vertices of a regular pentagon centered at the origin. The eigenvalues for these games are similar, but not identical, since the sampled population is not entirely regular. The Inferred Graph in the rightmost column plots a representative point for each of the pure strategies. Advantage relations that are not accounted for in the first disc game are represented in the second. In the first disc game the sequence of pure strategies, Rock, Paper, Scissor, Lizard, Spock, are spaced by $144°$ degrees about the pentagon. In the second, the sequential pure strategies are spaced $72°$ apart, so occupy adjacent corners of the pentagon.

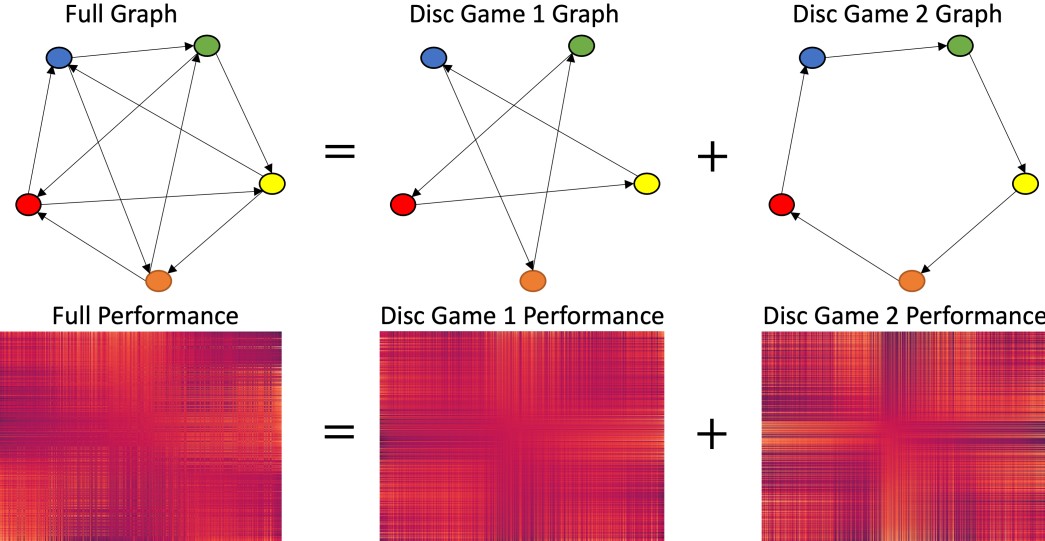

Figure 7: The first row shows the abstracted performance graphs for the full game as well as each disc game. Pure strategies are represented by color (red for rock, blue for paper, green for scissors, yellow for lizard, orange for spock). Arrows represent dominance relations. For example, the arrow from green to red in the middle column denotes the strong advantage rock possesses over scissors due to the first disc game. The heat maps in the second row represent performance matrices for the full game and for each disc game. The full performance relation is a sum of the two advantage cycles produced by each disc game.

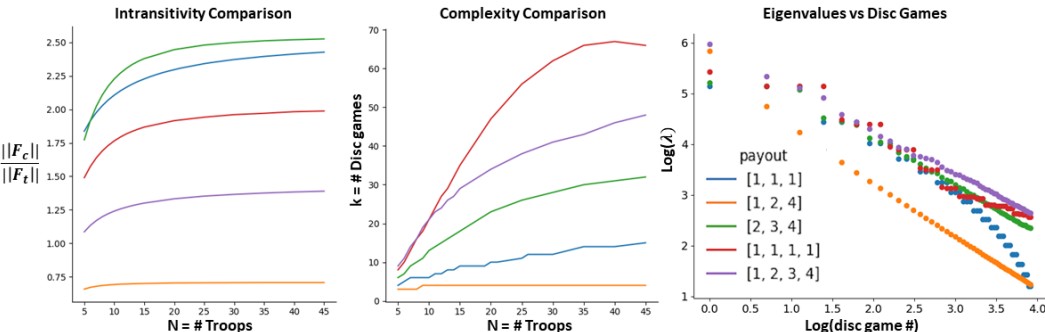

Figure 8: For a variety different blotto games we have **Left:** $\frac{\|F_c\|}{\|F_t\|}$ vs allocation (troops) $N$ **Middle:** complexity vs $N$ where complexity is the number of disc games $k$ it takes to reach $0.05\%$ Frobenius norm error rate, **Right:** For each game we list the eigenvalues in decreasing order using log scale

To clarify the advantage relations encoded by each disc game, we use the inferred graph to construct an abstract representation of a performance graph for the pure strategies. The result is shown in Figure 7. Colors denote strategy, and arrows denote advantage. The full game graph can be decomposed into the two separate game graphs embedded in each of the disc games.

# E    BLOTTO: INTRANSITIVITY AND COMPLEXITY ANALYSIS

Here we look at how PTA can be useful for comparing different blotto games. New blotto games are easily created by varying the game parameters. The number of zones, troops, and payouts can be changed to generate games with varying structure. Other Blotto variants, such as Boolean Blotto, or Colonel Lotto, are not considered here. We focus on a small number of zones — $K = 3$ or $K = 4$

— to illustrate how the intransitivity ($\frac{\|F_c\|}{\|F_t\|}$) and complexity change for varying $N$ and $K$ across a number of different payout structures.

We propose three hypotheses. First, while the number of distinct strategies grows combinatorially in $N$ and $K$, the allotment problem converges, in the limit of large $N$, to a continuous problem in which each commander can allot an arbitrary fraction of their total force to any zone. Therefore, our structural measures should converge to finite values representing the structure of a Blotto game allowing any fractional unit allotment on the interval $[0, 1]$. Both measures should increase with increasing $N$ towards the limiting case, as the space of available allotment strategies grows with $N$. Second, complexity should increase with increasing $K$, since the number of distinct strategic trade-offs should increase as the number of distinct, exploitable, win conditions increases. Third, small changes in game structure can lead to large differences in both intransitivity and complexity. Small changes can break underlying symmetries, and, the majority functions that determine performance are discontinuous in nearby allocations so small changes in battlefield weights can produce sudden changes in the set of win conditions.

To test these hypotheses, we vary $N$ between 5 and 45 while holding $K$ fixed at 3 and 4. We consider three distinct payout structures for K=3 and 2 for K=4. For each game we construct $F$ and compute the intransitivity $\frac{\|F_c\|}{\|F_t\|}$ using the HHD. The complexity of each game is computed by finding the number of disc games required to reach an error tolerance of $0.05$ (measured in relative Frobenius error). All of the games considered are highly cyclic, with $\|F_c\| > \|F_t\|$ in all but the $[1, 2, 4]$ case. The $[1, 2, 4]$ case is transitive case, since the only win condition is victory in. Note that $\|F_c\|$ is not zero for the $[1, 2, 4]$ case since the chosen performance measure cannot be expressed as a linear function of a difference in ratings, so is not perfectly transitive. Nevertheless, if the step function used to assign battlefield outcomes is replaced with any sigmoid $s(x)$, then the $[1, 2, 4]$ case is perfectly transitive with respect to $s^{-1}$, where rating equals allotment to zone 4.

Figure 8 shows the results, and confirms our three hypotheses. First, the intransitivity and complexity for all games plateaus for large enough N. This indicates that, beyond a certain $N$, the game reaches its "strategic capacity"; all meaningful types of trade-offs have been expressed. In both cases, the limiting complexity is strikingly small relative to the size of the strategy space, which grows rapidly in $N$. For a game with $K$ zones and $N$ units, there are $(N + K - 1$ choose $K - 1) = \mathcal{O}(N^K)$ distinct strategies to consider. In the most extreme case tested, $N = 45$ and $K = 4$, so the strategy space contains $\mathcal{O}(10^4)$ distinct allotments, which can be reduced to $\mathcal{O}(10)$ distinct trade-offs. Thus PTA can effectively separate the underlying complexity of a game from the size of its strategy space.

Second, in the right side of Figure 8 the complexity of the $K$=4 games are much higher than the $K$=3 games.

Third, the difference in complexity from one game to the next in stark. At $N = 45$ for example there are about 15 additional disc games needed to reach 95 % accuracy when going from the [1,2,4] transitive game to the uniform [1,1,1] game. Furthermore having similar $\frac{\|F_c\|}{\|F_t\|}$ does not imply similar complexity. The [1,1,1] and [2,3,4] games have similar intransitivity but have a difference of 20 in complexity.

## F  POKEMON

Here we describe in further detail the construction of the Pokemon data. In the full game a player (or trainer) captures Pokemon to compete against the Pokemon of other players, usually with teams of 6 Pokemon chosen at the player's discretion. Players also choose the order in which their Pokemon compete since the actual combat is done pairwise. This pairwise interaction is what allowed us to ignore the team aspect and still learn important aspects of the game. Each of the pokemon had a set of attributes. The attributes are shown below.

1. Type 1: Main Type - Fire, Water, Grass, ect...

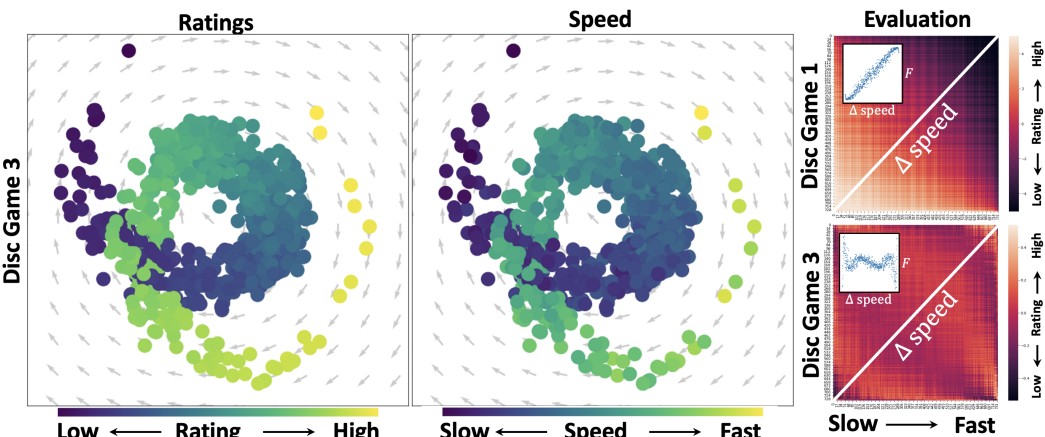

Figure 9: **Left:** Disc game 3 colored by rating. **Middle:** Disc game 3 colored by speed. **Right:** Evaluation matrices generated by disc games 1 (top) and 3 (bottom), with agents ordered by increasing speed. Both evaluation matrices are close to Toeplitz, so produce evaluation that depend primarily on the difference in speed between agents. The function that returns performance given a speed difference is approximated by sampling the evalutation matrix along the cross-diagonal marked in white. The subpanels containing scatter plots show the sampled evaluations.

2. Type 2: Secondary Type - Not all pokemon have two types but we did not find this to contribute to any performance tradeoffs in a significant way

3. HP: Hit points - Indicated how much damage a pokemon can endure before losing the match.

4. Attack: Base modifier for normal attacks

5. Defense: The base damage resistance against normal attacks

6. Special attack

7. Special Defense

8. Speed: This stat largely determines which pokemon get to attack first. As combat is turn based, this constitues a large advantage which we saw in disc game 1.

There were 50,000 pairwise interactions among the 735 pokemon that were used. The data for each interaction consisted of the name of the first and second pokemon as well as the winner of the match. In an individual matchup, each Pokemon has a certain level of HP or health. The two Pokemon take turns attacking one another until one of the them loses all of their HP and is declared the loser. The first to attack is determined by some set of attributes that is not explicitly given by the data set, but speed is known to be a large contributing factor. Since we did not have the full interaction graphWe filled in any missing data using logistic regression, producing a win probability matrix. We obtained the evaluation matrix $F$ via the logistic link function commonly used in Elo rating. The evaluation for competitor $i$ vs competitor $j$ is then given by $f_{i,j} = \log(\frac{p_{ij}}{1-p_{ij}})$ where $p_{ij}$ is the probability that Pokemon $i$ beats Pokemon $j$. We applied the Schur decomposition directly to $F$ to show that disc game embedding can successfully isolate a dominant transitive component (speed).

In Figure 9 we show the third disc game left out of the main analysis. It shows a double loop structure with a full inner circle and a half outer circle. Like disc game 1, disc game 3 is, approximately, a curve parameterized by speed. As in the Fourier examples discussed before, the double loop represents a higher order correction to disc game 1. Disc game 1 confers a transitive, monotonically increasing advantage to faster agents. The faster an agent relative to their opponent, the larger their advantage. Disc game 3 adds nuance to this relation by discounting the advantage conferred by small differences in speed, increasing the advantage conferred by intermediate differences in speed, discounting the advantage conferred by large speed differences, and strongly rewarding maximal

speed differences (see the evaluation matrix and associated subpanel in the rightmost column of Figure 9).

Note, these corrections to disc game 1 are very small. The eigenvalue for disc game 1 is roughly 15 times larger than the eigenvalue for disc game 3, hence the relationship between speed and performance is largely determined by disc game 1. We did not discuss disc game 3 in the main text for this reason.

