# OpenReview forum: "Principal Trade-off Analysis"
_ICLR.cc/2023/Conference — Submitted to ICLR 2023_

### Official Review · Reviewer_1u6d · 2022-10-23

**Confidence:** 3
**Correctness:** 4
**Technical Novelty And Significance:** 4
**Empirical Novelty And Significance:** 4
**Recommendation:** 8

**Clarity, Quality, Novelty And Reproducibility:**

The paper is very well written, and the idea is very novel. I include more details in the main review.

Nitpick: use \citep to get parenthetical citations.

**Strength And Weaknesses:**

The concept of applying a PCA-like analysis to zero-sum games is very intriguing and, to my knowledge, novel. The method presented is simple yet powerful, and creates very interesting analyses of games. I think this is a good contribution to the literature, and as such I vote to accept.

One potential avenue of improvement is to include explicitly-worked decompositions of simple zero-sum games. For example, what is the decomposition of generalized RPS (i.e., 2k+1 actions of each player; action i beats action j if (i-j) mod (2k+1) <= k) or generalized matching pennies (normal-form game, n actions per player, payoff = identity matrix)? (of course, properly symmetrized.) That may give some intuition for what the method is doing--for example, we know exactly what the equilibria should be in both cases (uniform random).

I have only a few minor questions:

1. Why decompose $F_c$ instead of directly decomposing $F$ itself? That is, why ever explicitly separate out the transitive component? The decomposition could also be used on $F$ itself, and the transitive component would naturally appear as a component of the decomposition if it is prominent enough. This is what happened in the Pokemon example, and it produced a nice-looking plot of the transitive component. It would be interesting, in my opinion, to see that component explicitly in the Blotto example.

1. Does this method have any implications for *solving* games? For example, if a straightforward method exists to solve games with low rank, one could apply that method to a low-rank approximation of a game that is otherwise too complex to be solved, and hopefully recover a good strategy in that game as well.

1. How dependent are the embeddings $\vec y_k(i)$ on the particular choice of (low-dimensional) embedding of the strategy space? For example, in the example Blotto game, one could hypothetically simply represent each pure strategy $i$ as a basis vector $\vec e_i \in \mathbb{R}^{2926}$. Would that result in different visualizations? How different?

Post-response: Thank you for the detailed response. I thought before that this was a strong paper, and I still do. I maintain my score.


**Summary Of The Paper:**

The authors introduce principal trade-off analysis, which is in some sense an analogue of PCA for zero-sum games. They propose an algorithm for computing PTA, and run it on several games of interest, producing informative visualizations.

**Summary Of The Review:**

A solid paper, introducing a new and seemingly very powerful method of analyzing two-player zero-sum games. I vote to accept.

---

> ### Author Response · Authors · 2022-11-19
> **Response to Strength And Weaknesses and to minor question 1**
>
> ### Explicitly worked example response
>
> We appreciate the value of simple introductory examples and have revised the text accordingly. We started with Blotto since it was simple to present and complex enough to admit an illuminating disc game representation. Simpler games were presented in past cited work [Balduzzi et al. 2019]. Since we aim to illustrate the explanatory power of PTA, we need an example whose representation was non-obvious a priori, but was interpretable once found.
>
> That said, we agree that it is important to build intuition regarding the representation so that a reader can understand the more nuanced examples. We have updated the manuscript to include a RPS+2 example in the appendix. Please see our response to reviewer 3 for a detailed description of the changes.
>
> ### 1. Why decompose the cyclic component:
>
> This is a valid question, and one we have debated internally. In principle, we see both approaches as reasonable methods. The method used should depend on the user’s desired representation.
>
> On one hand, the transitive component is equivalent to a comparison of ratings, so does not need a two-dimensional disc game representation. We raise this point when describing our method in the second paragraph of section 4:
>
> “The transitive component can be represented on a line via the ratings, so does not require additional visualization.”
>
> If the user is interested in treating the two components separately, then they should decompose F, and adopt the simplest representation of each component.  Adopting this procedure amounts to representing the game as a combination of two parts. The first (transitive) is a comparison of agent rating/quality, so does not need PTA. The second (cyclic) is a combination of trade-offs, so demands PTA. This was the approach proposed in [Balduzzi et al. 2018b].
>
> On the other hand, if the user is interested in representing the entire game with PTA, then they should Schur decompose F. If the transitive part is large enough, then it is apparent in the resulting visualization. By applying PTA to all of F at once, the user can represent the full game via a combination of trade-offs, rather than the cyclic component alone. This was, in practice, the method we adopted in our experiments.
>
> Our presentation on this issue was not clear, since the methods section described the first approach, while the experiments used the second. We have revised the methods section to describe the second approach as the main method, and to suggest the first approach as a possible modification in the case when a user intends to view the transitive and cyclic components separately.
>
> Please see the added paragraph in section 4:
>
> “A user interested in the transitive and cyclic components of F separately, could begin by breaking F into F_t and F_c. The transitive component can be represented on a line via the ratings, so does not require additional visualization. The cyclic component F_c is skew symmetric, so can be represented via PTA. Then, performance is represented by a combination of two components. The first compares the overall quality of the agents, as quantified by a set of ratings. The second represents any cyclic relations as a combination of principal trade-offs. We apply PTA to F, not F_c in all of our experiments.”
>
> ### Sources for the response:
>
> Balduzzi, David, et al. "Open-ended learning in symmetric zero-sum games." International Conference on Machine Learning. PMLR, 2019.
>
> Balduzzi, David, et al. "Re-evaluating evaluation." Advances in Neural Information Processing Systems 31 (2018).
>
> Adsul, Bharat, et al. "Rank-1 bimatrix games: a homeomorphism and a polynomial time algorithm." Proceedings of the forty-third annual ACM symposium on Theory of computing. 2011.
>
> Kannan, Ravi, and Thorsten Theobald. "Games of fixed rank: A hierarchy of bimatrix games." Economic Theory 42.1 (2010): 157-173.
>
> Barman, Siddharth, et al. "The empirical implications of rank in bimatrix games." Proceedings of the fourteenth ACM conference on Electronic commerce. 2013.
>
> Monga, Amnol, and Quanyan Zhu. "On solving large-scale low-rank zero-sum security games of incomplete information." 2016 IEEE International Workshop on Information Forensics and Security (WIFS). IEEE, 2016.
>
> Lipton, Richard J., Evangelos Markakis, and Aranyak Mehta. "Playing large games using simple strategies." Proceedings of the 4th ACM Conference on Electronic Commerce. 2003.

---

> > ### Comment · Reviewer_1u6d · 2022-11-19
> > **Thank you**
> >
> > Thank you for the detailed response. I thought before that this was a strong paper, and I still do. I maintain my score.
> >
> > One reason I asked about the the transitive component is that not all transitive games are the same. For example, the disk game whose strategy set is an origin-centered arc (of less than 180 degrees) is not the same as the disk game whose strategy set is a line segment, though both are transitive. So the visualization of the transitive component may still be valuable.

---

> ### Author Response · Authors · 2022-11-19
> **Response to minor question 2 and 3**
>
> ### 2. Solving games:
>
> This is an interesting question, and a topic we are actively investigating. The complexity of finding Nash Equilibria (mixed or pure) is well studied. There are some results in the literature that suggest that, when payout matrices are low-rank, Nash equilibria, or approximations to Nash equilibria, can be found efficiently (in polynomial time instead of in exponential time). For example, given a general 2 player bimatrix game, mixed Nash Equilibria in zero sum and rank-one games can be found in polynomial time [Adsul et al. 2011]. In rank-k games, ϵ approximate equilibrium can be found in poly(1/ϵ) time [Kannan and Theobald 2010]. Low rank approximation is also relevant in empirical settings when player choices are observed but payouts are unknown and must be inferred [Barman et al. 2013], or when the payout matrices are not fully known [Monga and Quanyan. 2016].
>
> As an example, Lipton, Markakis, and Mehta [Lipton et al. 2003] showed that all normal form games (m player) admit epsilon-Nash equilibria with small support (logarithmic in the number of pure strategies), where the mixed strategy distribution is uniform over a small subset of the possible strategies. Such approximations can be found in quasi-polynomial time. This approach is attractive since it is both computationally efficient, and produces simple strategies that are easy to implement. Moreover, if the payout matrices are low rank, then there exist exact equilibria with small support. Consequently, if a payout matrix is close to low rank, then low rank approximations to the payout matrices admit equilibria with small support that can be computed in quasi-polynomial time. Those equilibria are epsilon NE for the original game. Thus, low rank approximation offers cheap, simple, approximations to Nash equilibria. We have added a sentence to the text in the third paragraph of section 4 to emphasize this result.
>
> We are also interested in using PTA and the associated mapping into disc games to guide training protocols. For inspiration, see [Balduzzi et al. 2019]. We are actively developing training methods that seek ensembles of diverse, effective agents, by perturbatively expanding the convex hull of the embedded population. We hope to describe these efforts in future work.
>
> ### 3. Dependency on the strategy space representation:
>
> The embeddings are entirely independent of the trait space representation since it only depends on agent performance and uses no information regarding agent parameterization. The embedding acts pointwise, mapping from a specific agent to a specific location in a series of disc games. Hence the disc game embeddings only represent information about the advantage relationships among the individual players.
>
> That said, the relation between trait space and embedding is important and subtle. Our examples clearly demonstrate that there is a meaningful mapping from agent attributes to locations in disc game space. If we aimed to recover a functional embedding, that mapped explicitly from agent traits to agent locations in disc game space, then the parameterization of the trait space matters since, to recover a functional embedding, we need to restrict the class of allowed embedding functions. Once a function class is chosen, the choice of trait space parameterization matters.
>
> Similarly, the density of agents sampled is implicitly related to their parameterization. For example, assuming a uniform distribution of action probabilities across agents would not be the same as assuming a uniform distribution of weights in a neural net determining action probabilities. Each case would produce a different population of agents. The population of agents used does influence the embedding since our approach is empirical. PTA summarizes a game as it is played by a specific population. Changing the background distribution of agents would change the spectral representation of the game used for embedding.
>
> While these are important problems, they are beyond the scope of the current work and will be addressed in a forthcoming paper in preparation.
>
> ### From clarity section. Use \citep to get parathetical citations:
>
> Thank you for this suggestion. We have updated the manuscript accordingly.
>
> Please see list of sources on other comment

---

### Official Review · Reviewer_pi9j · 2022-10-24

**Confidence:** 2
**Clarity, Quality, Novelty And Reproducibility:** See Summary Of The Review
**Correctness:** 3
**Technical Novelty And Significance:** 3
**Empirical Novelty And Significance:** 3
**Recommendation:** 3

**Strength And Weaknesses:**

Strengths:

- Understanding the game structure rather than focusing on the equilibrium computation.

- PTA reduction of a game in simple cyclic games.


Weaknesses:

- Lack of clarity.

**Summary Of The Paper:**

The authors study the overall structure of games. They propose the Principle Trade-off Analysis (PTA) a decomposition of two-player zero-sum game. Similar to Principal Component Analysis PTA represents a game as the weighted sum of pairs of orthogonal plane. Where each plane is associated with a strategic trade-off from a simple cyclic game. Furthermore truncation to the first planes provides a meaningful reduction of the game. The authors illustrate the usefulness of the method on two games Blotto and Pokemon. In particular in  Blotto allows to recover specific symmetries whereas in Pokemon PTA recovers clusters that naturally correspond to Pokemon
types and correctly identifies a strategic trade-off in the Pokemon generation type.

**Summary Of The Review:**

#Review
Understanding the overall structure of a game rather that focusing only on computing equilibrium is a interesting research direction. And in particular the provided decomposition method is a valuable contribution. But the paper lacks of clarity in order to fully present the PTA method. In particular I find the presentation a bit too verbose and it could be improved with supporting precise quantitative statement and definition of the introduced quantities. It would be also useful to illustrate the different visual properties on a very simple example before jumping to the Blotto and Pokemon games. Finally the literature section could be improve by explaining what are the other methods of reductions, compare them to PTA in the experimental section.


#Specific comments:

- P2, related work: It is not very clear at this point what are the other methods and what are the differences between them and PTA.

- P3, Section 3.1: Can you detail the Helmholtz-Hodge decomposition and precise what is 'least squares ratings that evaluate the average performance of each agent'.

- P3, Section 3.1: Often we do not have access to the exact advantage F but rather samples, how does you  model can be adapted to this case?

- P3 Section 4: 'Each pair of consecutive columns' of which matrix?

- P3, Section 4: define SVD.

- P3, Section 4: It'll be easier to follow if you define precisely what are F_t and F_c.

- P3, (2): Ok but you already used the notations Q and U for the decomposition of F when you said you replaced A by F.

- P3 below (2):  Can you define what is the optimal rank 2r approximation for F_c.

- P4, Section 4: Actually you never defined properly what is PTA.

- P4, above (4): Add a comma for the lower index of F_{c,i,j}.

- P4, last paragraphs: Can you  details these properties quantitatively?

- P5, last paragraph: How do you get discrete allocation from the sample of a Dirichlet?

- P6, Blotto Example 1: At this point it is hard to understand Table 1, especially since some notations used in it are still not introduced.

- P6: What is exactly a trade-off? Is it the same thing as a disc-game?

- P6, exchange symmetry: I'm not sure I follow this part. Why such subtlety is not  treated in the previous section presenting PTA? And since generic games should not exhibit such strong symmetries why choosing this game as first example?

- P7 Second paragraph: At this point it is not completely clear for me what you plot in Figure 1. Can you detail precisely which quantity in function of what is plotted.

- P8, Section 5.2: What are the agents in this case?

---

> ### Author Response · Authors · 2022-11-18
> **Response to Summary of the review and first few comments**
>
> ### Summary of the review
> We hope that our responses to your specific comments will make the paper clearer and address any confusion regarding definitions. We agree with that it would be beneficial to illustrate PTA on a simpler example. We used blotto because it is simple enough for illustration but complex enough to demonstrate utility. We have added an additional example to the appendix that is simpler.  We hope it helps to clarify the visuals used in the main text. We did not incorporate the example into the main text since the other reviewers all noted the clarity and effectiveness of our presentation, simple examples are presented elsewhere, and there is not space to provide three examples. Nevertheless, we have revised the text to try and address the specific concerns you raised, and hope the revised version is easier to follow.
>
> ### P2, related work
> Many other approaches such as ones we mention in the related work section take the approach of trying to learn a high level abstraction of the game. This could be seen as a reduction. They then use this to help inform training dynamics [Omidshafiei et al. 2020; Garnelo et al. 2021] or to help inform ranking in games with cyclic structure [Omidshafiei et al. 2019]. PTA however finds trade-offs that explain performance in an embedded space (disc games). PTA therefore takes a much more fine grained approach to the previous mentioned ones. In order to get a reasonable comparison we could use PTA to infer graphical structures of the game that could be used for training. See the RPS+2 example in the appendix for an idea of how this might be done. Our work as builds on [Balduzzi et al. 2018] by emphasizing the interpretability of the representation. We are unaware of other work that takes a similar enough approach to provide a baseline comparison.
>
> ### P3, Section 3.1
>
> The Helmoltz Hodge Decomposition is described in detail in the papers referenced in P3.1. Please see The network HHD: Quantifying cyclic competition in trait-performance models of tournaments https://doi.org/10.1137/20M1321012 for a complete description. The basic idea follows:
>
> Any performance matrix F can be broken into a transitive component F_t and a cyclic component F_c. The transitive component between two agents, i and j equals the difference in a rating assigned to each agent. That is, f_(t_ij )=r_i-r_j. These ratings are the solution to a least squares problem, which seeks the ratings r such that F_t is the best approximation to F in Frobenius norm. The necessary ratings satisfy a simple recursive statement. The rating of agent i equals their average performance against their neighbors, plus the average rating of their neighbors. When F is complete (all agents compete with all other agents), the rating of agent i equals their average performance against all other agents. Once the ratings are found, the transitive component is fixed, so the cyclic component can be recovered by setting F_c  = F - F_t. This approach is well documented in the sources listed above and is not the focus of this paper. It is enough for the reader to know that such a decomposition is possible.
>
> We have revised the methods section to work primarily with F, since our focus is on PTA, not on the HHD.
>
> ### Sources:
>
> Stewart, Gilbert W. Perturbation theory for the singular value decomposition. 1998.
>
> Balduzzi, David, et al. "Re-evaluating evaluation." Advances in Neural Information Processing Systems 31 (2018).
>
> Omidshafiei, Shayegan, Christos Papadimitriou, Georgios Piliouras, Karl Tuyls, Mark Rowland, Jean-Baptiste Lespiau, Wojciech M. Czarnecki, Marc Lanctot, Julien Perolat, and Remi Munos. "α-rank: Multi-agent evaluation by evolution." Scientific reports 9, no. 1 (2019): 1-29.
>
> Shayegan Omidshafiei, Karl Tuyls, Wojciech M Czarnecki, Francisco C Santos, Mark Rowland,Jerome Connor, Daniel Hennes, Paul Muller, Julien Pérolat, Bart De Vylder, et al. Navigating thelandscape of multiplayer games. Nature communications, 11(1):1–17, 2020
>
> Marta Garnelo, .. Pick your battles: Interaction graphs as population-
> level objectives for strategic diversity. arXiv preprint arXiv:2110.04041, 2021.

---

> ### Author Response · Authors · 2022-11-18
> **Continued Response to specific comments**
>
> ### P3, Section 3.1
>
> We agree with the reviewer that the true performance function is rarely known. Indeed, we assume that the entries in F simply represent the observed performance between two competitors. It may or may not come from the true performance function. Pokemon for example was not constructed from the exact advantage F.
>
> For most games we will not have access to the true performance function so will be restricted to the empirical setting used here. PTA does not require complete knowledge of F at all possible inputs, since it is well defined for any sample population. That said, the stability of the method to errors in estimating F (as when payout is defined as an expected payout from a stochastic game and is estimated from observed games), or to changes in the population, is an important issue. The stability of the embedding under errors in the entries may be studied using the standard perturbation analyses of spectral methods - namely, perturbations of the singular value decomposition. See, for example [Gilbert 1998].
>
> ### P3, Section 4
> Each pair of consecutive columns of Q, the orthonormal basis used in the Schur decomposition of A. We specified the matrix in the sentence in the parenthetical immediately followed the quoted remark:
>
> “Each pair of consecutive columns, [q_(2k-1),q_2k], …”
>
> We have modified the sentence to read:
>
> “Each pair of consecutive columns of Q, [q_(2k-1),q_2k], …”
>
> ### P3, Section 4
>
> SVD is the Singular Value Decomposition. We have replaced SVD with singular value decomposition in the text.
>
> ### P3, Section 4
>
> F_t and F_c refer to the transitive and cyclic components of F found using the HHD as described in section 3.1. We modified section 3.1 to provide clearer definitions.
>
> ### P3 (2)
>
> We introduced Q and U earlier when we said “any skew-symmetric matrix A admits a Schur decomposition (real Schur form), QUQ^T” in the first sentence of Section 4. This was intended as a general statement about skew-symmetric matrices, of which F is a special case.
>
> ### P3, Below (2)
>
> Here by 2r optimal we mean using the first 2r eigenvectors from Q and the first 2r× 2r upper block from U. It is optimal in the sense that it gives the smallest approximation error between the original matrix F and the one reconstructed using the first 2r components. We have modified the text to more clearly explain this definition of optimality.
>
> ### P4, Section 4
>
> Thank you for this comment. We have revised the manuscript to include a specific definition of PTA at the start of section 4. The first sentence in section 4 now reads:
>
> “PTA decomposes an arbitrary performance matrix F into a sum of simpler performance matrices by embedding each agent into a series of disc games that model important strategic trade-offs. “
>
> Equation 4 makes this description explicit and precise. Please see Appendix B for more details.
>
> ### P4, commas between indices
>
> In response to reviewer 4’s second comment we have updated the methods section, replacing F_c with F. There is now no need for stacked subscripts. We do not believe a comma is needed between pairs of indices specifying a matrix entry, though we are happy to add the comma if required.
>
> ### P4, last paragraphs
>
> All properties described in the first half of the paragraph are quantitative and are standard properties of the cross product. The interpretation in terms of advantage is special to disc games but is meant to build intuition for the visuals. We have rewritten the last paragraph of section 4 using equations to try and clarify what was described in words.
>
> ### P5, last paragraph
>
> For Blotto, we use the same number of α values as there are battlefields and then set them all to one to ensure an even sampling of the strategy space. To force all the samples to be valid strategies, convert to closest integer values, then check whether their sum is greater than the number of allowed units. For implementation details, please see the “make_strat” function within blotto.py provided in the supplementary code.
>
> Please see first comment for sources

---

> ### Author Response · Authors · 2022-11-19
> **Response to remaining specific comments**
>
> ### P6, Blotto Example 1
>
> We apologize that the table was not fully explained. We have added a sentence to the caption defining the abbreviations H, M, and L used. To our reading, all other terms are defined. The type labels are new to the table but are shared by both table 1 and figure 1 and are defined in the figure caption. We also provide a column containing representative allocations that provide examples of each type. The abbreviation D.G. is defined in the caption, and the simplex is defined in the caption as the simplex of possible allocations. Please point us to other terms, if any, that were not clear.
>
> ### P6, what is a trade-off
>
> Mathematically, a trade-off is an advantage relationship that can be represented by embedding into a disc game. The conceptual justification for this definition follows from the structure of the cross-product, which models a canonical cyclic performance relation between two attributes.
>
> While PTA factors advantage relationships into independent tradeoffs, it does not tell us what these trade-offs represent in terms of traits of players or other features of the game. In this respect PTA, is similar to PCA, where the principal directions may represent complex combinations of primitive features. We demonstrate here, how, by visualizing the embeddings, it is possible to relate trade-offs observed in performance back to trade-offs in the trait space. For example, table 1 lists the strategic interpretation of the trade-offs identified in Blotto.
>
> ### P6, exchange symmetry
>
> We did not raise this subtlety earlier since it is only likely to occur in very select cases with strong symmetries. These cases occur in constructed examples where mathematical simplicity induces symmetry.
>
> We raise this point for two reasons. First, it is a limitation of the method not addressed in previous work. This is not reason enough, since, as mentioned, it is a limitation that only applies to special cases. Second, it is essential to understand the implications of exchange symmetry in order to follow the Blotto example. We chose Blotto as our first example since it is both simple enough to explain and reason with, but also complex enough to admit a nontrivial embedding. Simpler games do not offer this compromise and are illustrated in other sources. Since we aim to show that PTA can recover strategic features, we need an example where the strategic features are not apparent a priori but can be interpreted once discovered. Blotto offers clearly interpretable tradeoffs that are elegantly geometric, easily understood, and non-obvious before analysis. It is also a useful example since it illustrates limitations of the method, both regarding symmetry, and regarding the interpretability of higher order trade-offs.
>
> That said, we appreciate the need and value of simple examples that build intuition. To that end, we have added a new RPS+2 example to the appendix, and point reader’s seeking a simpler example to it in the first paragraph of section 5.
>
> ### P7 Second paragraph
>
> Figure 1 plots the first 3 disc games from the blotto 1 example. Each row corresponds to a disc game. Each scatter cloud is a series of embedded agents. For example the first row uses coordinates $[y_{1} (i),y_{2} (i)]$ for each i, where Y is defined by equation (3). Each column colors the agents by a different attribute of the agents. The first column colors by agent rating. The next three columns color by the amount each agent allocates to a given zone. The last two columns show the angle and radius assigned to each agent. The location within the simplex of each scatter points represents the allocation used by that agent. The color of the scatter point represents either the radius assigned to each agent, or the angle, in the corresponding disc game embedding.
>
> ### P8 what are the agents?
>
> The agents here refer to the various pokemon as described in the opening sections of 5.2. Agents in this case are the pokemon themselves. Please see section E of the appendix for a more complete description.

---

### Official Review · Reviewer_d7g3 · 2022-10-25

**Confidence:** 3
**Clarity, Quality, Novelty And Reproducibility:** The paper is written in a clear way. …
**Correctness:** 4
**Technical Novelty And Significance:** 3
**Empirical Novelty And Significance:** 3
**Recommendation:** 6

**Strength And Weaknesses:**

Good visualization and coherent writing. I particularly liked the game balancing concept introduced and how we can visualize it through disk games. Similar insights are valuable for game understanding, beyond the utility function. It would be even more interesting to depart from the consideration of the reconstruction of the matrix Fc with a given accuracy and focus more on how to maximally utilize the information given by the first k disc games and capture elements of the game without the need of visualization. Higher dimensions wouldn’t allow visualization of the disc games.

Questions:
Given a reconstruction accuracy delta, how many disc games are needed to extract the useful information presented by the experiments? (such as the attack matrix). How does one argue about the redundancy of disc games in higher dimensions when visualization and understanding by humans is not possible?

**Summary Of The Paper:**

This paper introduces a method of decomposing games in low dimensional feature spaces called PTA. This method allows some characterization of game structure and allows for a general technique for visualizing data arising from competitive tasks or pairwise choice tasks. The relationships between embedding coordinates represent performance relations and elucidate the principal trade-offs responsible for cyclic competition in each game.


**Summary Of The Review:**

The visualization aspect of disc games is insightful. However, the results are mostly empirical. Formulating the PCA analogy for game structures allows the reconstruction of the matrix Fc with minimal disc game information. However, as the authors state, the difficulty in many games is creating the cyclic component Fc altogether. It would be interesting to answer how someone can optimally use the first k linear combinations of disc games without necessarily requiring visual feedback to understand.

---

> ### Author Response · Authors · 2022-11-18
> **Response to Visualization in higher dimensions, maximally utilizing information and how many disc games are needed.**
>
> Thank you for your thoughtful review. There are a few points that we would like to clarify.
>
> ### 1. Response to Visualization in higher dimensions
>
> PTA decomposes tournament data into a representation as a series of orthogonal 2D disc games representations. Each disc game is 2D regardless of the dimensionality of the game/tournament data, so visualization of the disc games is possible for any game.
> That said, visualization in the trait space itself, as performed in the Blotto example (angle and radii plots), is not possible in higher dimension. Moreover, the visual strategies we employ here (coloring by specific attributes), are harder to use in high dimensional trait spaces where it is less obvious how to color, and where much more work is needed to manually inspect diagrams. Then, automatic methods relating recovered embeddings back to interpretable trade-offs are needed. Possible methods include fitting an embedding to a function class with regularizing terms that promote sparse dependence on the input traits as in sparse PCA (see, for example [Zou et al. 2006]). Automated tradeoff interpretation should follow demonstration that PTA produces meaningful tradeoffs. We aim to accomplish the latter first.
>
> ### 2.	Maximally utilize information
>
> Maximally utilizing the information present in a disc game depends on what is meant by utilize. Utilize for what task?
> Our goal in this paper was to show that the disc game representation can be used to extract meaningful strategic features of the underlying game – as experienced by a chosen sample population. We are actively discussing other applications of the representation. We saved that discussion for future work since it is essential to first establish that PTA produces a meaningful representation of the game, and since our attempts to utilize the representation for other tasks are still works in progress. The existing work which introduced the disc game representation used it to better understand population training, diverse team selection, and as a latent space to design and classify games (see the referenced works from Balduzzi’s group at Google DeepMind). We are investigating classification methods which classify games based on PTA and training methods that seek diverse populations of agents that independently exploit each trade-off identified by PTA.
> We have modified the conclusions paragraph to include suggestions towards these future directions:
> “Future work could provide more general methods for finding embeddings, such as a functional theory connecting performance with attribute space, could seek a sparser representation via extensions of sparse PCA [Zou et al. 2006]. Future work should also investigate automated methods that summarize the trade-offs identified by PTA without the need for visual inspection, and that leverage the representation for game classification, construction, and exploration.”
>
> ### 3.	How many disc games are needed?
>
> As in PCA, the number of significant disc games is equal to the number of significant eigenvalues. Specifically, consider the sum of squares of the magnitudes of the eigenvalues. Given a desired accuracy tolerance, use the smallest number n of disc games needed such that the square root of the sum of squares of the first n eigenvalues divided by the Frobenius norm of F equals or exceeds the tolerance.
> This point was explained in the second to last paragraph of section 4. We have copied the relevant material here for your reference:
> “Nevertheless, the sequence of embeddings form optimal low rank approximations to F, where the importance of each embedding is quantified by the associated eigenvalue. Thus, the sequence of eigenvalues determines the number of disc game embeddings, r, required to achieve a sufficiently accurate approximation of F.”
>
> Sources for response:
>
> Zou, Hui, Trevor Hastie, and Robert Tibshirani. "Sparse principal component analysis." Journal of computational and graphical statistics 15.2 (2006): 265-286.

---

> > ### Comment · Reviewer_d7g3 · 2022-11-28
> > **Thank you for your response**
> >
> > Dear authors and AC,
> >
> > I am really sorry for the late reply. I am satisfied with the responses of the authors and decided to increase my score from 5 to 6.
> >
> > Best,
> > Reviewer

---

> ### Author Response · Authors · 2022-11-18
> **Response to Redundancy and Difficulty in creating the cyclic component.**
>
> ### 4.	On redundancy:
>
> All disc game embeddings produced by PTA are independent in the sense that, if an agent is chosen uniformly from the population, then each embedding coordinate is independent of each other embedding coordinate. In this sense, the embeddings are never redundant, regardless the dimension. We raise this point in section 4. We have copied the relevant text below for reference:
> “In PTA, the embeddings are projections onto orthogonal planes, so each embedding encodes independent information about cyclic competition. Here independence means that the embedding coordinates of a randomly sampled agent in two distinct planes are uncorrelated. Note that this definition depends on the density of sampled agents.”
> That said, it is not always clear whether enforcing independence via orthogonality produces meaningfully distinct strategic features. This is a subtle point, and one we raise when discussing the Blotto examples. In particular, see the last paragraph of section 5.1 which reads:
> “Strikingly, subsequent disc games imitate the disc game 1-3 trade-offs, only at higher frequency on a smaller spatial scale in allocation. This suggests that the disc games may act like Fourier modes, where early disc games capture low frequency, global trade-offs, and later disc games capture high frequency, local trade-offs. It also suggests that orthogonality may not be the appropriate notion of independence for trade-offs. A sharper notion of equivalency is needed. Methods like nonnegative matrix factorization, which address similar issues among PCA features, suggest an avenue for further development.”
> We are actively exploring the use of sign constraints, or sparsity promoting regularization, that may lead to more clearly interpretable trade-offs. In these cases the appropriate definition of “redundant” varies depending on what is desired in the trade-offs. One reasonable notion is that two disc games are independent if it is possible to select a location in one without constraining location in another. This notion is much stricter than orthogonality and does not allow as direct a decomposition method.
>
> ### 5.	Difficulty in creating cyclic component
>
> The methods described for decomposing a performance matrix into transitive and a cyclic matrix are founded on the Helmholtz-Hodge Decomposition (HHD). The HHD is computationally efficient for large problems since it requires solving a sparse least squares problem. When working with a complete matrix F where all the entries are known, the HHD has cost O(n^2) where n is the number of agents.
> As the reviewer suggests, the largest computational cost is associated with the collection of the performance data forming F. In the examples given, we work with a full performance matrix, and with an exhaustive collection of agents spanning all possible strategies. Simulating all pairwise interactions, and producing all agents, is expensive in large games. This expense can be addressed through adaptive sampling and matrix completion methods. Our initial tests indicate that a small percentage of matches are needed to reliably reconstruct the disc games in certain examples. Preliminary tests using the Pokemon data showed that only 5 - 10% of the matrix entries were needed if PTA was augmented with low-rank matrix completion. This suggests the possibility of working with far fewer simulations, so may allow much more efficient computation. Moreover, we are working on training methods, that iteratively expand a population based on observed tradeoffs. We hope that such methods may allow efficient exploration in games with large strategy spaces.
> We raised the possibility of using matrix completion in paragraph 6 of section 4. We have modified the end of the paragraph to read:
> “The simulation cost could be reduced if low-rank completion methods were applied to fill in missing data. We leave sampling considerations and matrix completion methods to future work.”

---

### Official Review · Reviewer_enae · 2022-10-25

**Confidence:** 3
**Correctness:** 4
**Technical Novelty And Significance:** 4
**Empirical Novelty And Significance:** 3
**Recommendation:** 8

**Clarity, Quality, Novelty And Reproducibility:**

The writing is high-quality, though the typesetting could definitely be improved (see above for detailed comments). There are no obvious reproducibility concerns.

**Strength And Weaknesses:**

I really enjoyed the paper. To use the same wording as the authors, my research mostly focuses on the "policy problem", not on the "problem problem", so my point of view is that of a non-expert in the specific research niche of the paper. I found the writing style engaging, never boring, very approachable, and exceptionally persuasive.

For those reasons, assuming that none of the reviewers that are expert on the topic have issues with the positioning (e.g., all relevant literature has been correctly and fairly represented) and claims of the paper, I recommend acceptance of the paper.

That being said, I think that the authors should revise the paper to improve certain aspects:
- throughout the paper, a majority of the citations should have been contained in parentheses and instead are just part of the text; this is rather distracting. Please make sure you're using the \citep and \citet commands adequately.
- many plots have axes whose tick labels are straight up unreadable.
- many (all?) plots are raster images, rather than vectorial graphic. Please consider re-exporting all your plots to be vectorial.
- the bibliography section seems to be using the wrong style file. This is the only paper I'm reviewing that has underlined venues in the bibliography. Please align the bibliographic style to the rules of the conference.
- some of the bibliographic entries have wrong capitalizations: for example, I believe Go and Blotto should be spelt with capitals, to name two examples

**Summary Of The Paper:**

The paper suggests using a representation similar to PCA to highlight important tradeoffs associated with different winning conditions in arbitrary games in a way that lends itself to 2D visualization.

**Summary Of The Review:**

Overall, I appreciated the high-quality writing and persuasiveness of the paper. I don't have any particular concern with the paper, or reason to believe the proposed method has obvious shortcomings that would prevent it from being useful beyond the games considered in the paper. The typesetting quality needs to be improved for the final version.

---

> ### Author Response · Authors · 2022-11-18
> **Strengths Weakness and Revision**
>
> Thank you for your kind review. We apologize for the typesetting and styling issues. We have updated the manuscript with the necessary corrections. In regards to the plots, you are right that the tick labels were not readable. We have updated Figure 1 and Figure 2 to address this concern. We considered changing the plots to vectorial graphic as you suggested but we think that after making our most recent adjustments to the plots we could not get any additional benefits from a vectorial graphic format.

---

### Decision · Program_Chairs · 2023-01-20

**Decision:**

Reject

**Justification For Why Not Higher Score:**

There are absolutely no evidence that this approach  can be generalized, apart from 2 use-case quite specific.

**Justification For Why Not Lower Score:**

N/A

**Metareview: Summary, Strengths And Weaknesses:**

This paper looks at 0-sum game that are skew-symmetric (which is *not* compulsory, unlike the authors say). Using the Schor decomposition, it is possible to "replace" the original payoff matrix by a sum of $k$ "simpler" ones (as in PCA, hence the terminology).

Then this idea is illustrated on two different games, a variant of Blotto and another game related to the Pokemon franchise.

Honestly, the pictures are beautiful. But that's it. There is not much gain of intuition in any of those games, and the choice to focus on those two is quite disputable.

If there is a real interest of this approach, then it should be demonstrated properly (and not just verbosely as one reviewer mentions) on more than two (maybe) well-chosen games. For instance, how does the equilibria of the sub- game relate to the equilibria of the original one ? I wanted to ask how the value of the game is impacted, but since the construction works for skew-symmetric game only, this is irrelevant. Can it be generalized, somehow, to a larger class of games ?

I have the feeling that there might be something behind this idea, but it has not been investigating enough to be certain of it.